# Trial summary and protocol for a phase II randomised placebo-controlled double-blinded trial of Interleukin 1 blockade in Acute Severe Colitis: the IASO trial

Martin Geoffrey Thomas,[1] Carrie Bayliss,[1] Simon Bond,[1,2] Francis Dowling,[1] James Galea,[3] Vipul Jairath,[4,5] Christopher Lamb,[6,7] Christopher Probert,[8] Elizabeth Timperley-Preece,[9] Alastair Watson,[10] Lynne Whitehead,[11] John G Williams,[12] Miles Parkes,[13,14] Arthur Kaser,[13,14] Tim Raine[13,14]

For numbered affiliations see end of article.

**Correspondence to**
Dr Martin Geoffrey Thomas;
martin.thomas@addenbrookes.nhs.uk

## ABSTRACT

**Introduction** Acute severe ulcerative colitis (ASUC) is a severe manifestation of ulcerative colitis (UC) that warrants hospitalisation. Despite significant advances in therapeutic options for UC and in the medical management of steroid-refractory ASUC, the initial treatment paradigm has not changed since 1955 and is based on the use of intravenous corticosteroids. This treatment is successful in approximately 50% of patients but failure of this and subsequent medical therapy still occurs, with colectomy rates of up to 40% reported. The Interleukin 1 (IL-1) blockade in Acute Severe Colitis (IASO) trial aims to investigate whether antagonism of IL-1 signalling using anakinra in addition to intravenous corticosteroid treatment can improve outcomes in patients with ASUC.

**Methods and analysis** IASO is a phase II, multicentre, two-arm (parallel group), randomised (1:1), placebo-controlled, double-blinded trial of short-duration anakinra in ASUC. Its primary outcome will be the incidence of medical (eg, infliximab/ciclosporin) or surgical rescue therapy (colectomy) within 10 days following the commencement of intravenous corticosteroid therapy. Secondary outcomes will include disease activity, time to clinical response, time to rescue therapy, colectomy incidence by day 98 post intravenous corticosteroids and safety. The trial aims to recruit 214 patients across 20 sites in the UK.

**Ethics and dissemination** The trial has received approval from the Cambridge Central Research Ethics Committee (Ref: 17/EE/0347), the Health Research Authority (Ref: 201505) and Clinical Trials Authorisation from the Medicines and Healthcare products Regulatory Agency. We plan to present trial findings at scientific conferences and publish in high-impact peer-reviewed journals.

**Trial registration number** ISRCTN43717130; EudraCT 2017-001389-10.

## Strengths and limitations of this study

► A multicentre, UK-wide trial of 214 participants at 20 sites.
► The largest first-line treatment study ever conducted in this patient group.
► Potential for a step change and improved outcomes in patients with acute illness, including cost savings for the National Health Service.
► Pragmatic trial with wide eligibility criteria.

diarrhoea, rectal bleeding and abdominal pain. UC affects around 120 000 people in the UK.[1] UC tends to follow a relapsing/remitting course with periods of comparative health interspersed with unpredictable acute 'flares'. These may be severe enough to mandate hospital admission. For patients with UC in the UK, annual hospitalisation rates with acute severe UC (ASUC) are around 3% (3600 individuals). Twenty to forty per cent of these patients will require emergency surgical removal of the colon (colectomy) during the same hospital admission.[2 3] A recent UK audit of emergency surgery in colitis showed mortality rates of 4% at 90 days.[4] Additionally, surgery carries significant morbidity associated with the need for at least a temporary stoma, as well as long-term complications (both physical (eg, adhesional disease) and psychological) and complications of subsequent procedures (eg, pelvic nerve damage with ileal pouch-anal anastomosis or completion proctectomy).

Treatment for ASUC involves high-dose intravenous corticosteroids and is standardised worldwide.[5] After 3–5 days, patients who do not exhibit an improvement may

## INTRODUCTION
### Background and rationale
Ulcerative colitis (UC) is an inflammatory condition of the colon associated with

be considered for medical 'rescue' therapy with further immunosuppression using either infliximab or ciclosporin. Those patients who subsequently fail to respond to medical treatment, or patients with fulminant colitis at presentation, will then proceed to surgery, typically within the space of 4–10 days of rescue therapy initiation. This is usually performed on the same admission with a subtotal colectomy and creation of an end ileostomy.

The initial management of ASUC has changed little since the seminal trial of intravenous corticosteroids in 1955.[6] Despite significant advances in therapeutic options and drug development for ambulatory patients with UC, patients with ASUC are typically excluded from the eligibility criteria of standard clinical trials for patients with moderate to severe UC. Clinical trials that have compared infliximab or ciclosporin as medical rescue treatments for ASUC have required patients to fail to respond to treatment with up to 5 days of intravenous steroids.[7–9] In the absence of reliable early prognostic markers, it may be appealing to treat all patients presenting with ASUC with immunosuppressive treatments normally reserved for rescue therapy rather than corticosteroids alone. The barriers for this approach include the toxicity of such treatment[10–13] as well as the financial burden incurred.[14] There is still therefore an unmet need to improve the management of ASUC by doing something at the point of initial treatment, rather than waiting for treatment failure.

One of the key mediators of colonic inflammation in ASUC is interleukin 1 (IL-1) which plays a pivotal role in local activation of neutrophils and a number of downstream inflammatory mediators.[15] The IL-1 axis has been repeatedly identified as a therapeutic target in UC.[16–18] IL-1 antagonism occurs naturally via the ubiquitous IL-1 receptor antagonist (IL-1Ra). A recombinant form of this naturally occurring anti-inflammatory, anakinra (Kineret), has been used for the treatment of patients with rheumatoid arthritis (RA) and cryopyrin-associated periodic syndromes (CAPS) with good effect.[19–22] Anakinra has a very good safety profile in many disease states including RA, CAPS, severe sepsis and intracranial haemorrhage at standard subcutaneous dosing regimens and using high-dose intravenous administration regimens.[21 23 24]

Our hypothesis is that antagonism of the IL-1 axis with anakinra is an effective way of abrogating the early inflammatory response seen in ASUC and leads to improved clinical outcomes.

There is evidence to suggest that anakinra cotreatment with corticosteroids may be beneficial, given that corticosteroids suppress the production of IL-1Ra to a greater extent than its active family members—IL-1α and IL-1β.[25] Moreover, while recent research has seen a trend towards antagonism of IL-1 using monoclonal antibodies,[17] anakinra may offer some advantages. First, its relative short half-life (4–6 hours) compared with antibodies allows early institution of further immunosuppression therapy if rescue therapy is required. Second, anakinra antagonises both IL-1α and IL-1β, while monoclonal antibodies are specific to either form. Finally, given that the patent for anakinra is now expired, the cost of the product is much less than that of the monoclonal antibody alternatives.[26 27] Given this evidence, antagonism of IL-1 early in the course of inflammation, in conjunction with standard corticosteroid treatment, is a promising therapeutic avenue in patients with ASUC.

### Aims and objectives
The Interleukin 1 blockade in Acute Severe Colitis (IASO) trial aims to test whether anakinra can reduce the need for medical or surgical rescue therapy in patients with ASUC when given alongside intravenous corticosteroids. The trial also aims to assess the effects of anakinra on disease biology and bowel inflammation in addition to assessing drug safety, longer term health and patient-reported quality of life outcomes. Finally, the trial will seek to advance scientific understanding of ASUC through the collection and analysis of a large range of biological samples, including blood and tissue transcriptomic analysis, as well as microbial analysis. As part of this work, we will attempt to validate a transcriptional predictor of outcomes in UC that we have previously reported.[28]

### Primary objective
To compare the clinical effects of anakinra with placebo when given in addition to current standard care in patients with ASUC.

### Secondary objective
To compare the effects of anakinra with placebo when given in addition to current standard care to patients with ASUC in terms of:
► Safety.
► Need for colectomy.
► Patient-reported outcomes.
► Endoscopic and histological evidence of treatment effects.
► Evidence of disruption of the IL-1 signalling pathway.

This article summarises the approved trial protocol in use at the time of submission (version 2.0, dated 25 May 2018). The full version of the protocol can be found in the online supplementary material 1. The most current version of the protocol will be made available online at https://www.journalslibrary.nihr.ac.uk/programmes/eme/1420102/#/.

## METHODS
### Trial design and flow chart
IASO is a multicentre, two-arm (parallel group), randomised, placebo-controlled, double-blinded trial. The trial will also feature a substudy specifically examining the endoscopic effects of treatment with anakinra in comparison with placebo. A summary of the trial design is shown in figure 1.

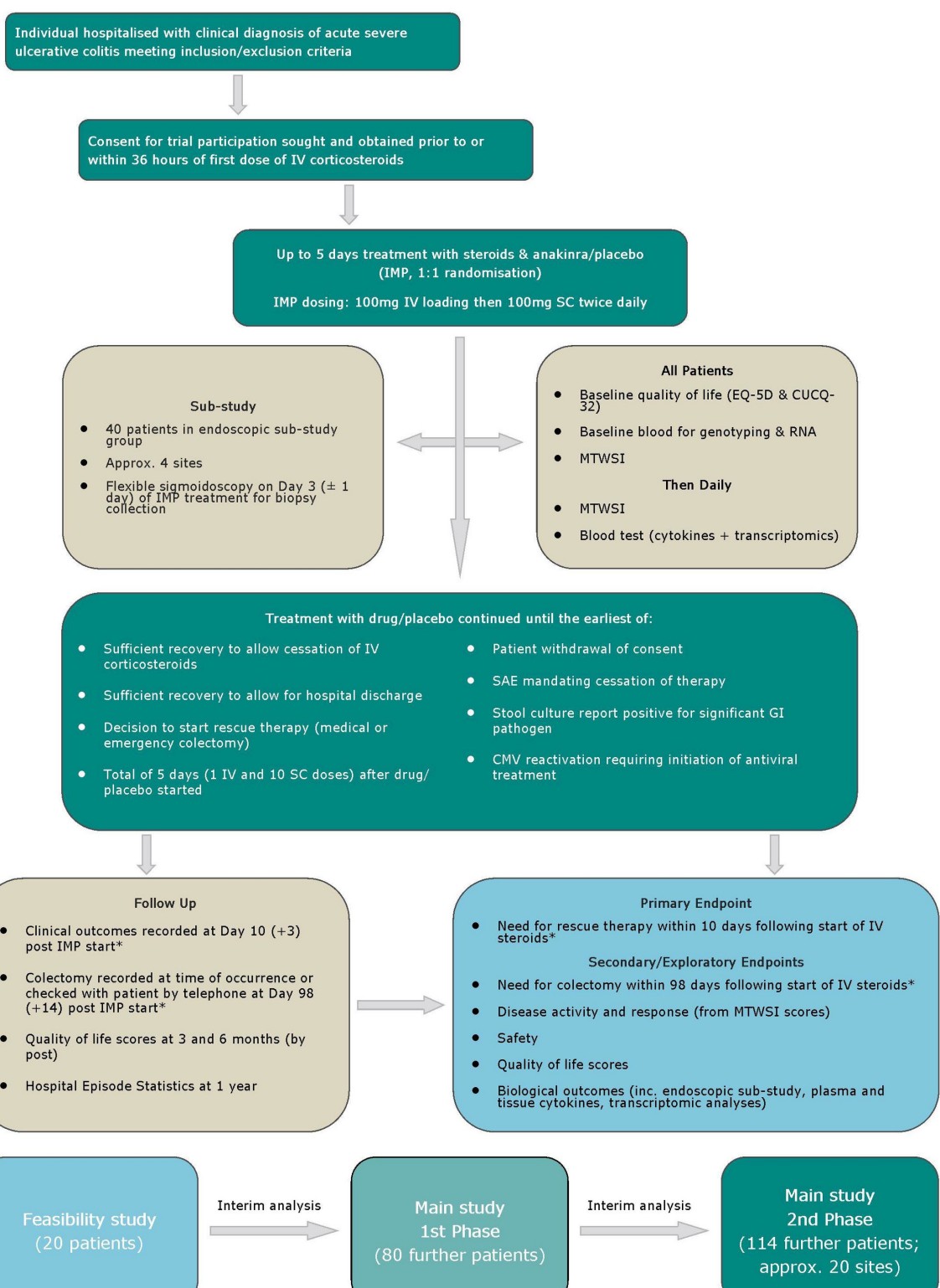

**Figure 1** Interleukin 1 blockade in Acute Severe Colitis (IASO) trial flow chart. *Data collection for the primary endpoint and the colectomy-specific secondary endpoint is performed at days measured relative to the start of IMP treatment. However, the subsequent data analysis will assess these endpoints using time relative to the earlier start of intravenous corticosteroid treatment. CMV, cytomegalovirus; CUCQ-32, Crohn's and Ulcerative Colitis Questionnaire-32; EQ-5D, EuroQol five dimensions; GI, gastrointestinal; IMP, investigational medicinal product; IV, intravenous; MTWSI, modified Truelove and Witts severity index; SAE, serious adverse event; SC, subcutaneous.

## Centres

IASO is a multicentre study in UK National Health Service (NHS) acute hospitals. We plan to include approximately 20 centres in the trial.

All sites will recruit, treat and follow-up participants equally, but a subset of sites will also recruit participants into the optional endoscopic substudy involving one additional flexible sigmoidoscopy with biopsy collection.

## Trial duration

Participants' involvement in the trial will last approximately 6 months. This includes an initial treatment period of approximately 5–7 days, during which screening and an investigational medicinal product (IMP) treatment course of up to 5 days will be completed. Review of participant outcomes following trial participation may continue for up to 5 years using Hospital Episode Statistics (HES) data, without any additional burden to the participant. Any HES data collection would start at 1 year after commencement of IMP treatment and examination of outcomes via HES data would potentially continue for up to 5 years, depending on study outcomes. A yearly assessment of HES outcomes by the Trial Steering Committee (TSC) would determine whether collection of HES data should continue.

## Participant selection
### Inclusion criteria
To be included in the trial the participant must:
► Be aged 16–80 years inclusive.
► Have given written informed consent to participate.
► Be hospitalised with clinically confirmed or suspected ASUC and a modified Truelove and Witts severity index (MTWSI)[7] score ≥11.
► Have a requirement for treatment with intravenous corticosteroids in the judgement of the treating clinician, with the possibility to receive a first dose of IMP within 36 hours of commencement of intravenous corticosteroids.

There are no validated clinical scoring systems for ASUC. We selected the MTWSI (also known as the Lichtiger index) with a cut-off of ≥11 since this aligns with the score used in index studies of ciclosporin in ASUC,[7] as well as in the CycloSporine versus InFliximab (CySIF), and Comparison of Infliximab and Ciclosporin in Steroid Resistant Ulcerative Colitis (CONSTRUCT) trials comparing ciclosporin and infliximab.[9 29] The components of the MTWSI include those parameters that are relevant to clinical decision-making for initial recognition of ASUC and for determination of the need for rescue therapy (see online supplementary material 2).

### Exclusion criteria
The presence of any of the following will preclude participant inclusion:
► Pregnant or breastfeeding women.
► Oral corticosteroid dosing for a duration of 8 weeks or more immediately prior to commencement of intravenous corticosteroid dosing.
► History of severe hepatic impairment (eg, Child-Pugh grade C).
► Moderate or severe renal impairment (estimated glomerular filtration rate (eGFR) <60 mL/min/1.73 m$^2$).
► Neutropenia (neutrophil count <1.5×10$^9$/L).
► Previous treatment with anakinra for any indication.
► Documented hypersensitivity to the active substance or to any of the excipients or to *Escherichia coli*-derived proteins; latex allergy.
► Evidence (eg, blood cultures) or clinical suspicion of systemic infection (Concurrent prescription of antibiotics to cover for the possibility of gastrointestinal (GI) infection while awaiting stool culture results, or the possibility of bacterial translocation relating to severe colitis, is not an exclusion criterion where the physician suspects UC is the most likely diagnosis).
► Current or previous cytomegalovirus (CMV) infection requiring treatment with antiviral agents.
► Current treatment with anti-tumour necrosis factor (TNF)-α therapy or anti-TNF-α discontinuation within previous 16 weeks.
► A history of pulmonary tuberculosis infection.
► Any absolute contraindication to intravenous corticosteroid.
► History of malignancy (with the exception of non-melanoma skin cancer) or colonic dysplasia.
► Rectal therapy in previous 14 days (substudy exclusion only).
► Receipt of another IMP as part of a Clinical Trial of an Investigational Medicinal Product within the previous 16 weeks.

## Outcome measures
### Primary outcome
The primary outcome of the trial will be the incidence of medical (ie, infliximab or ciclosporin) or surgical rescue therapy (colectomy) within 10 days following the commencement of intravenous corticosteroid therapy.

For the purposes of the primary endpoint analysis, the time point of the start of rescue therapy will be as follows:
► Medical rescue therapy: date and approximate time of first treatment administration.
► Surgical rescue therapy: date and approximate time of start of surgery.

### Secondary outcomes
1. Incidence of colectomy within 98 days following commencement of intravenous corticosteroid therapy.
2. Burden of disease activity, measured by daily MTWSI scores over Days 1–5 after initial IMP administration.
3. Time to clinical response (defined as second consecutive day with MTWSI <10) (Time to clinical response is measured during the IMP treatment phase only).

4. Time to medical or surgical rescue therapy, measured according to the time after the first dose of intravenous corticosteroids until the time that rescue therapy occurs (using definitions as set out in the primary endpoint). Data will be captured up to the same time point as the primary endpoint.

5. Incidence of serious adverse events (SAE), measured until Day 10 (+3) following commencement of IMP treatment.

### Exploratory outcomes

1. Patient-reported quality of life (EuroQol five dimensions, (EQ-5D)[30] and Crohn's and Ulcerative Colitis Questionnaire-32 (CUCQ-32)[31]) at baseline and at 3 and 6 months following commencement of IMP treatment.

   The EQ-5D is a widely used generic health-related quality of life measure featuring self-assessment questions distributed across five domains. The CUCQ-32 is an inflammatory bowel disease (IBD)-specific questionnaire which captures information related to a patient's well-being during their daily routine. Both questionnaires have been validated.[32 33]

2. Endoscopic response at Day 3 (±1 day) (via standardised endoscopic scoring completed by a local and a central blinded assessor) following commencement of IMP treatment (endoscopic substudy group only).

In addition to the above exploratory endpoints, the trial will also perform further exploratory analyses related to the identification of biomarkers and treatment response predictors, as well as characterisation of the stool microbial response to anakinra. Finally, patient readmission and colectomy data (eg, via HES) may be examined to investigate the impact of treatment in the longer term.

### Trial treatment

For the purpose of this trial, the following are considered the IMP:

► Anakinra solution for injection.
► Placebo to match anakinra solution for injection.

IMP is supplied in commercial prefilled syringe configurations (100 mg/0.67 mL), packed into trial-specific participant kits containing 11 prefilled syringes. The syringes and kits have blinded labelling and are identified by unique pack (kit) numbers. IMP will be stored in accordance with the requirements of the Summary of Product Characteristics for Anakinra.[34]

Participants will receive an initial loading dose of 100 mg IMP given intravenously within 36 hours of first intravenous corticosteroids. This will be followed by 100 mg subcutaneous IMP given twice daily, over a maximum of 5 days (maximum of 10 doses).

Intravenous methylprednisolone or hydrocortisone, as per local guidelines for ASUC standard care, will be administered to participants as non-IMPs. Participants will continue to be treated with routine concomitant medications for other comorbidities as appropriate within their current clinical condition, with modifications (eg, oral to intravenous switches), or temporary cessation as necessary. Appropriate supportive care may also be prescribed, including fluids, nutritional supplements and other medications as necessary.

Participants will return to their normal standard of care as defined by their local physicians, following completion of trial treatment.

### Treatment termination and trial withdrawal

Participants may choose to withdraw from treatment or the full study at any stage without prejudice to standard clinical care.

Clinicians should cease treatment with the study drug according to their judgement or under any of the following criteria:

► Withdrawal of consent for treatment administration.
► Development of renal impairment (eGFR <60 mL/min/1.73 m$^2$) or severe hepatic impairment (Child-Pugh grade C).
► Development of neutropenia (neutrophil count <1.5×10$^9$/L).
► Detection of a significant GI pathogen in stool specimen.
► Sufficient recovery to allow for cessation of intravenous steroids.
► Hospital discharge due to recovery.
► Decision to commence medical or surgical rescue therapy.
► CMV reactivation necessitating treatment with antiviral agents (in the judgement of treating clinicians).
► The development of an SAE necessitating termination of IMP treatment.

Where possible, trial assessments will continue following treatment termination.

In addition to the treatment termination criteria, the following criteria would result in participant withdrawal from the trial, with no further assessments being performed:

► Withdrawal of consent for further assessments and data collection.
► Death.

### Randomisation and stratification

All patients screened for the trial will be assigned a unique participant ID number. Suitable participants will subsequently be randomised (1:1) to drug:placebo using an online randomisation system accessible via password-protected access. Randomisation will be stratified for two variables:

► Previous or current therapy with any of: immunomodulators (azathioprine, 6-mercaptopurine, 6-thioguanine, methotrexate, ciclosporin), biologics (anti-TNF-α monoclonal antibodies, anti-adhesion molecule antibodies, other anti-cytokine antibodies) or oral Janus kinase inhibitors.
► Current or previous oral corticosteroid prescription(s) within 8 weeks prior to first dose of intravenous corticosteroids.

## Blinding

Trial participants, research teams and site pharmacies will be blinded to the treatment group for the duration of the trial.

## Procedures and assessments

### Participant identification and informed consent

We will recruit inpatients with ASUC. Identification of potential participants will be undertaken by the clinical team with subsequent consenting and trial-specific procedures carried out by the research team.

A total of 40 individuals will be asked to join an optional scientific substudy cohort. Participants can decline to participate in the substudy but still participate in the main study.

Anonymised data on all participants who are approached will be collated in accordance with Consolidated Standards of Reporting Trials guidelines.

### Screening

The tests required for screening all form part of standard clinical care, therefore no additional tests will be performed other than a urine β-human chorionic gonadotropin test for females where there is a possibility of pregnancy.

### Baseline

The following activities will be performed and recorded at baseline:

► Medical history review.
► Review of blood parameters (haemoglobin, total white cell, neutrophil, lymphocyte, monocyte, platelet, albumin and C-reactive protein) following a blood test performed as part of standard care.
► Three research blood samples (one plasma tube, one nucleic acid tube, one genotyping tube).
► Concomitant medications review.
► Demographics.
► MTWSI assessment.
► Stool sample collection.
► EQ-5D and CUCQ-32 questionnaires.

### Randomisation and post-baseline procedures

Following baseline data collection, participants will be randomised and allocated treatment as described above.

Prior to the first administration of IMP (Day 0), the following assessments will be performed:

► Review of adverse events (AE).
► Recording of any changes to concomitant medication since the last assessment.
► Confirmation that IMP administration can proceed, according to the following criteria:
 – The time of administration of the first IMP dose falls no later than 36 hours following the time of first dose of intravenous corticosteroid.
 – Intravenous corticosteroid therapy remains ongoing at the time of administration of the first IMP dose (ie, that a decision to discontinue corticosteroids or transition to oral steroids has not been

made after the time of participant consent to participate in the trial and before the time of the administration of the first IMP dose).
 – A firm decision to start rescue therapy or perform emergency colectomy has not already been taken.

During the days subsequent to the first dose of trial medication, the following activities will be recorded daily up to and including Day 5 following IMP treatment initiation:

► MTWSI assessment/review of standard care clinical MTWSI assessment domains (Not recorded for patients who have undergone colectomy).
► Recording of any changes to concomitant medication prescription since the last assessment.
► Review of blood parameters (haemoglobin, total white cell, neutrophil, lymphocyte, monocyte, platelet, albumin and C-reactive protein) following blood tests performed as part of standard care.
► Two research blood samples (one plasma tube, one nucleic acid tube) (A minimum of one sample every 72 hours should be taken. Wherever possible, every effort should be made to obtain the samples every day (up to 10 samples)).
► Review of AEs.
► Medical or surgical (emergency colectomy) rescue therapy assessment.
► Stool sample collection on Day 5 (±1 day) following commencement of IMP.

### Substudy-only assessments

In addition to the daily assessments described above for Days 1–5, the scientific substudy group will undergo an additional optional flexible sigmoidoscopy (without bowel preparation) on Day 3 (±1 day) following commencement of IMP. This will be performed during their index inpatient admission.

Findings will be recorded according to a standardised endoscopic scoring system (Mayo)[35] by local and central blinded assessors. Histology will be scored on up to a maximum of six biopsies taken from each of the rectum and sigmoid colon, according to validated scoring systems for colitis severity at the end of the trial.[36–38]

Where possible, endoscopic scores, written summary reports and photographic recordings (video or static images) of sigmoidoscopies will be sent to the central trial team for verification of local scoring. In addition, where available, data collected as part of standard care (non-substudy) endoscopies will be used to supplement trial-specific data. These will be scored by the blinded central assessor.

### Follow-up

Following the participants' inpatient stay, the following assessments will occur:

► Day 10 (+3) following commencement of IMP treatment—rescue therapy assessment (In the event that a participant is discharged prior to Day 10 without having started rescue therapy, a member of the trial

team will contact the participant on Day 10 (+3 days) to confirm that the participant remains out of hospital and has not received rescue therapy elsewhere since discharge. If the participant is not contactable, local hospital records will be checked for admission).

► Day 10 (+3) following commencement of IMP treatment—AE monitoring.

► Day 98 (+14) following commencement of IMP treatment—colectomy assessment (If not previously recorded, a member of the trial team will contact the participant on Day 98 (+14) to confirm that the participant remains out of hospital and has not undergone a colectomy since discharge. If the participant is not contactable, local hospital records will be checked for admission).

► Approximately 3 and 6 months following commencement of IMP treatment—EQ-5D and CUCQ-32 questionnaires.

Participant status following discharge, including readmissions and subsequent surgery, may be assessed using patient-specific HES data from the NHS Digital, Data Access Request Service. Any data collection would start at 1 year after commencement of IMP treatment and examination of outcomes via HES data would potentially continue for up to 5 years, depending on study outcomes.

### Safety monitoring and reporting

This section summarises the safety monitoring and reporting processes for the trial. Full details of the safety processes can be found in the comprehensive protocol included as part of the online supplementary material 1.

AEs will be recorded from the point of informed consent. Standard regulatory definitions for assessing AE seriousness will be used.[39]

As the IMP has a short half-life (<6 hours) no cumulative or late effects are anticipated. Therefore, recording of AEs and reporting of SAEs will be actively monitored up to Day 10 (+3 days). Following the end of the active monitoring period, investigators are still required to report any serious adverse reactions or suspected unexpected serious adverse reactions (SUSAR) of which they become aware.

Pregnancy reporting is only required for trial participants while they are receiving IMP and for up to 24 hours after their last dose.

### Patient and public involvement

Patient involvement was actively sought during the planning and preparation stage for this trial, and will form a key part of the trial as it progresses.

A 10-member patient steering group assisted with trial design in an early workshop, reviewed previous versions of the trial protocol and assisted with responses to the initial funding application review. The group provided valuable insight into the acceptability of different dosing regimens and into ways to support patients during the consent process. The patient information leaflet was reviewed by the group. To ensure that the views of the wider patient community were sought, an online questionnaire regarding willingness to participate in inpatient research studies in colitis was circulated to members of Crohn's and Colitis UK and responses used to inform trial design.

A patient representative will serve as a member of the TSC.

### Statistics and data management

#### Sample size

We plan to include approximately 214 participants (107 per group) in the trial with 40 participants also included in the optional endoscopic substudy. Recruitment of 214 patients to the main trial will give 85% power to detect a 20% absolute risk reduction (ARR) in the primary endpoint from a rate in the control group of 49% testing at the 5% significance level.

We have based our estimate of the primary endpoint rate on data from the recently completed National Institute for Health Research (NIHR)-funded CONSTRUCT trial.[40] This trial recruited 270 patients with ASUC across 52 UK sites between 2010 and 2013. In CONSTRUCT, 49% of patients admitted with ASUC needed escalation to medical rescue therapy following treatment with corticosteroids (Professor J Williams, personal communication). Since there are no existing estimates of effect size for anakinra in ASUC, we chose to power IASO to detect a 20% ARR following protocol review by members of the British Society of Gastroenterology IBD Clinical Research Group, who advised that this was the minimum effect size needed that would make a substantial difference to the patient population, and which would be broadly in line with effect sizes seen in other drugs for moderate to severe colitis.[41–43]

The scientific substudy based on testing cytokines will be performed on 40 participants. The choice of sample size is based on power studies in transcriptomic studies in UC[44] with adjustment for multiple testing, as well as pragmatic grounds of cost.

#### Patient population

Populations to be analysed in the trial are as follows:
► Full analysis population: all randomised participants who receive at least one dose of IMP.
► Randomised population: all randomised participants, regardless of whether IMP was received.
► Safety population: all consenting participants.

Any further populations may additionally be defined within the statistical analysis plan.

### Analyses

#### Primary analysis

We will report summary statistics on the primary and secondary endpoints according to treatment group. The primary analysis will consist of an estimate, 95% CI and p value of the absolute risk difference of the incidence rates of the need for medical or surgical rescue therapy within 10 days following the first administration of intravenous corticosteroids between the two treatment arms, using logistic

regression to adjust for important baseline covariates. The primary efficacy analysis will be based on the full analysis population.

The baseline characteristics will include:

► Prior diagnosis of IBD at the point of hospitalisation ('first presentation').
► Prior hospitalisation for ASUC.
► Previous or current therapy with any of: immunomodulators (azathioprine, 6-mercaptopurine, 6-thioguanine, methotrexate, ciclosporin), biologics (anti-TNF-α monoclonal antibodies, anti-adhesion molecule antibodies, other anti-cytokine antibodies) or oral Janus kinase inhibitors.
► Current or previous oral corticosteroid prescription(s) within 8 weeks prior to first dose of intravenous corticosteroids.
► Demographics.

Secondary endpoints will be compared in a similar regression model to estimate the treatment effect on an appropriate scale of comparison. This will include an assessment of the effects of treatment with anakinra on the secondary endpoints.

### Secondary analyses

The following secondary analyses will be performed in the trial:

► Analysis of the randomised population.
► Complier average causal effect analysis to assess the influence of the amount of treatment received, as distinct from treatment assigned.

To test the hypothesis that the clinical effects of anakinra in ASUC may differ between groups of patients with differing levels of prior inflammatory burden, prespecified subgroup analyses, in the form of estimating treatment-covariate interactions, will be performed on the following baseline characteristics:

► Prior diagnosis of IBD at the point of hospitalisation ('first presentation').
► Prior hospitalisation for ASUC.
► Naïvety to any of: immunomodulators (azathioprine, 6-mercaptopurine, 6-thioguanine, methotrexate, ciclosporin), biologics (anti-TNF-α monoclonal antibodies, anti-adhesion molecule antibodies, other anti-cytokine antibodies) or oral Janus kinase inhibitors at the point of hospitalisation.
► Receipt of oral corticosteroids within 8 weeks prior to first dose of intravenous corticosteroids.
► Cases of suspected or confirmed ASUC without evidence of CMV reactivation requiring treatment with antiviral agents and without evidence of a significant GI pathogen.
► Duration between first dose of intravenous corticosteroids and first dose of IMP.

### Interim analyses

As part of planned interim analyses, an independent Data Monitoring Committee (iDMC) will examine unblinded data and provide a recommendation to a TSC as to whether the trial should continue at two prespecified points detailed below.

After the first 20 participants have been recruited, and all completed a minimum of 10 days follow-up (to the time of the primary endpoint), trial feasibility will be assessed. The trial will be regarded as feasible only if all of the following measures are met:

1. ≥10% of eligible participants are recruited (based on use of screening logs to determine the percentage of eligible participants who are randomised in each site).
2. Adherence to dosing as per protocol (≥80% of participants received appropriately prescribed doses within any given 24 hours window).
3. Rescue therapy started on or before the seventh day following initial intravenous corticosteroid administration in ≥80% cases where required during the index admission (as determined by review of timing of rescue therapy administration relative to corticosteroid administration by trial investigators).
4. Maintenance of blinding: there should be no evidence of unnecessary unblinding of participants or investigators.
5. No serious safety concerns identified, especially with regard to infections or SUSARs.

After the first 100 participants have completed Day 10 assessments, futility analysis will be performed to test the hypothesis that the reduction in absolute risk difference in the primary outcome is ≥10% using a one-sided 2.5% significance test using logistic regression on the ARR scale. If the hypothesis is rejected, that is, the lower limit of a 95% two-sided CI for treatment—control is above −10%, then the iDMC will consider the recommendation for the study to stop early for futility. This is equivalent to spending 0.1% from the 15% total beta value set to control the type 2 error under the alternative hypothesis (49% and 29% rates).[45] If the study continues past the interim, then the final analysis will not be adjusted for the futility analysis, and standard p values and CIs will be reported.

### Data management

All outcome data will be transferred onto a case report form (CRF) which will be anonymised. Data captured on CRFs will subsequently be stored in a central database prior to analysis. Personal identifiable data kept for the purposes of questionnaire postage and HES data capture will be destroyed at the end of the trial.

### Missing data

Due to short and intensive period of follow-up for the primary endpoint, we do not anticipate significant missing data. Where we are unable to obtain missing data within 12 weeks of the 3 or 6-month time points, we will employ the standard approaches to management of missing quality of life data previously described in the CONSTRUCT trial (eg, participant death at questionnaire completion will be recorded as an EQ-5D score of 0

and match minimum observed scores for other quality of life domains).[40]

For participants who withdraw consent to treatment, permission to continue to acquire data for outcome analysis will be sought. For those who do not consent to ongoing monitoring, including those who wish to withdraw entirely from the study, existing data acquired to the point of withdrawal will be included in the final study analysis, with missing data handled according to standard missing data methods, including missing-at-random (MAR) methods with sensitivity testing for deviation from MAR assumptions.

## Trial end

For regulatory notification purposes, the end of trial will be the earliest of:

► The receipt of the final, returned 6-month quality of life questionnaires.
► Six months (+12 weeks) after the last participant entered the study (date of questionnaire censoring).

However, participants may enter a period of long-term follow-up via HES data examination and the trial would remain open to the Ethics Committee until the last participant's HES data capture has been completed.

## Ethics and dissemination

IASO has received ethical (17/EE/0347) and Health Research Authority approval. The trial is being carried out under a Clinical Trial Authorisation from the Medicines and Healthcare products Regulatory Agency.

This is an NIHR-funded trial and the NIHR publishing guidelines will be followed. Authorship of final study outputs will be assigned in accordance with guidelines set out by the International Committee of Medical Journal Editors. Plain language summaries of key trial outputs will be prepared in conjunction with the patient steering group and published on the trial website. This website is listed in the patient information leaflet for the benefit of all trial participants.

## Implications of pragmatic trial design

In designing the Interleukin 1 blockade in Acute Severe Colitis (IASO) trial, we have followed a 'pragmatic' approach to trial design. This aims to test our intervention under conditions as close to real-life routine practice as possible and so produce results that can be generalised and applied in routine practice settings. Another important aspect to consider was that, since the intervention targets an early molecule in the inflammatory cascade, and since there are already well-established options for rescue therapy, we wanted to test the intervention as close as possible to the point of recognition of a patient by a treating physician as having acute severe ulcerative colitis (ASUC). If the intervention is shown to have a positive outcome, then the generalisation of the trial findings might include consideration of initiation of anakinra

dosing at the same time as intravenous corticosteroids, that is, as part of initial therapy.

For these reasons, we aimed to keep our inclusion criteria as broad as possible and minimise exclusion criteria. Thus, in essence, almost any patient with ASUC capable of giving informed consent can participate in IASO. The key criterion is the recognition of ASUC by a physician and the judgement that intravenous corticosteroids are needed. The use of the modified Truelove and Witts severity index (MTWSI) serves as a 'sense-check' on this physician judgement to ensure that patients with too low a burden of inflammation or symptoms are not included. We considered and rejected other inclusion criteria that might be typically seen in 'explanatory' trials in inflammatory bowel disease (IBD), such as the use of video endoscopy with local or centralised scoring, or faecal calprotectin measurement. Although local assessment of the mucosa is widely used in the assessment of patients with ASUC, in many centres intravenous corticosteroids may be started before this point. Furthermore, the use of centralised scoring would introduce delays into participant screening to the extent that randomisation and dosing would be impossible to achieve prior to the point where rescue therapy should be considered. The use of physician judgement as the main arbiter for inclusion does risk the inclusion of patients who subsequently turn out to have, for example, Crohn's colitis or even infectious colitis, just as does happen in real life with decisions around corticosteroid initiation. We can, of course, obtain endoscopic data as well as clinical and histological data where available and these are indeed built into our analysis plans. But ultimately, IASO seeks to address whether a physician encountering a patient who he or she believes to have ASUC, on the basis of the evidence available, can and should prescribe anakinra in addition to corticosteroids. In this regard, more detailed inclusion criteria would not produce a generalisable result, while jeopardising timely patient identification and trial recruitment.

**Author affiliations**
[1]Cambridge Clinical Trials Unit, Cambridge, UK
[2]MRC Biostatistics Unit, Cambridge, UK
[3]University of Manchester, Manchester, UK
[4]Division of Gastroenterology, Department of Medicine, Western University, London, Ontario, Canada
[5]Division of Epidemiology and Biostatistics, Western University, London, Ontario, Canada
[6]Institute of Cellular Medicine, Newcastle University, Newcastle upon Tyne, UK
[7]Department of Gastroenterology, Newcastle Upon Tyne Hospitals NHS Foundation Trust, Newcastle Upon Tyne, UK
[8]Institute of Translational Medicine, University of Liverpool, Liverpool, UK
[9]Coventry, UK
[10]Department of Gastroenterology, University of East Anglia, Norwich, UK
[11]Clinical Trials Pharmacy Department, Addenbrooke's Hospital, Cambridge, UK
[12]Swansea University Medical School, Swansea, UK
[13]Division of Gastroenterology and Hepatology, Department of Medicine, University of Cambridge, Addenbrooke's Hospital, Cambridge Biomedical Campus, Cambridge, UK
[14]Department of Gastroenterology, Addenbrooke's Hospital, Cambridge University Hospitals NHS FoundationTrust, Cambridge, UK

**Acknowledgements** The authors thank the Cambridge Clinical Trials Unit for its support during the set-up and conduct of the trial. The authors also thank the patient steering group for its assistance with the trial design and for its ongoing involvement in the trial.

**Contributors** The full protocol was written by MGT, TR, AK, CB and FD. Sections related to statistics were written by SB. The trial treatment section was written by LW. The study design was conceived by TR with intellectual input provided by AK, MP, AW, SB, VJ, JGW, CP, JG, ETP and CL.

**Funding** The trial is sponsored by Cambridge University Hospitals NHS Foundation Trust and University of Cambridge. This project is funded by the Efficacy and Mechanism Evaluation (EME; 14/201/02) Programme, an MRC and NIHR partnership. (The EME Programme is funded by the MRC and NIHR, with contributions from the CSO in Scotland and NISCHR in Wales and the HSC R&D Division, Public Health Agency in Northern Ireland.) The trial drug is being provided by Swedish Orphan Biovitrum. The Wellcome Trust Sanger Institute is providing financial support and academic collaboration for sample sequencing. The Cambridge Clinical Trials Unit is supported by the National Institute for Health Research [Cambridge Biomedical Research Centre at the Cambridge University Hospitals NHS Foundation Trust]. CL is a clinical lecturer supported by the NIHR.

**Disclaimer** The views expressed in this publication are those of the author(s) and not necessarily those of the MRC, NHS, NIHR or the Department of Health and Social Care.

**Competing interests** None declared.

**Patient consent for publication** Not required.

**Ethics approval** Cambridge Central Research Ethics Committee (Ref: 17/EE/0347) and the Health Research Authority (Ref: 201505).

**Provenance and peer review** Not commissioned; externally peer reviewed.

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
