## [Reviewer comments · BMJ Open]

ARTICLE DETAILS

TITLE (PROVISIONAL)	Trial Summary & Protocol for a phase II randomised placebo controlled double blinded trial of Interleukin 1 blockade in Acute Severe Colitis – the IASO Trial
AUTHORS	Thomas, Martin; Bayliss, Carrie; Bond, Simon; Dowling, Francis; Galea, James; Jairath, Vipul; Lamb, Christopher; Probert, Chris; Timperley-Preece, Elizabeth; Watson, Alastair; Whitehead, Lynne; Williams, John; Parkes, M; Kaser, Arthur; Raine, Tim

VERSION 1 – REVIEW

REVIEWER	Dr Peter De Cruz The University of Melbourne Department of Medicine - Austin Academic Centre Melbourne, Victoria, Australia
REVIEW RETURNED	11-Jun-2018

GENERAL COMMENTS	This is a well-constructed pragmatic trial that addresses a neglected area of Acute Severe Ulcerative Colitis (ASUC) management ie how to reduce the need for medical and surgical rescue therapies which are well known to be associated with complications and morbidity. Specific comments The inclusion criteria are all clinically based - there are no endoscopic criteria required for entry. Primary endpoint of need for rescue/ Stats: 1. This assessment is based on the Oxford criteria and the presence of toxic megacolon and clinical judgment. The assessment of these criteria is open to interpretation and may be subject to site and investigator interpretation. Block randomisation by study site may help address this but it is not stated whether clock randomisation is part of the randomisation strata.2. The rescue rate of 49% in hospitalised patients is quite high (even though it is appreciated that this rate was derived from the CONSTRUCT recruitment data).3. The drop-out rate may be higher that expected due to the short screening and treatment window as exclusion criteria that are not apparent on screening may develop (eg bacterial gastroenteritis) Assessment
---

	The study would have benefited from video endoscopy as entry and assessment criteria to reduce cohort heterogeneity and improve the accuracy of outcome measure. General Comments and Safety aspects Whilst there are good mechanistic reasons why anakinra may work in active IBD there appears minimal pilot data to show efficacy. Preliminary data appear limited to a case report (Carter 2003) with worsening of CD and 3 cases of new onset UC after Anakinra (Hugle 2017). Anakinra is well tolerated in published literature; however, high serious infection rates have been found in RA literature in patients with concomitant steroids. This aspect of the IASO trial will therefore be particularly important to ascertain. Whilst the drug-half life may be short, there may be ongoing effects on the immune system ("immunological half-life") which may interact with infliximab or cyclosporin rescue therapies. The safety analysis of the first 20 patients will therefore be particularly important.
--	--

REVIEWER	Siddharth Singh University of California San Diego, La Jolla, California, USA
REVIEW RETURNED	21-Jun-2018

GENERAL COMMENTS	This is an interesting trial protocol on an important topic, exploring the addition of Anakinra to conventional IV corticosteroids in patients hospitalized with acute severe ulcerative colitis (ASUC). I have a few suggestions for the investigators' consideration:  1. In the abstract, please clarify what phase trial is this? PICO's need to be more clearly defined. Please also clarify that the intention is to use Anakinra for a short duration. 2. Please detail out MTWS index - is it a validated index for UC? Suggest using alternative indices too to facilitate comparison across trials such as the Lichtiger index? 3. Is endoscopic confirmation of disease activity a requirement for inclusion, for all patients? 4. What tests would be undertaken to rule out infection - GI pathogen panels based on nucleic acid amplification have 30% positivity, as compared conventional stool cultures? What is considered a "significant GI pathogen"? 5. I think it would be helpful to standardize IV corticosteroid course across centers to allow results to be interpretable 6. I am unsure about including patients who present with newly diagnosed UC patients treated with oral corticosteroids - what is proof of failing outpatient steroids? 7. I'm unsure but is concomitant cyclosporine allowed?? 8. If the goal is to understand 3 month outcomes, then I strongly suggest ascertaining treatment patterns after discharge, besides colectomy? When will physicians be unblinded? Will that influence subsequent therapy? 9. If possible, suggest measuring fecal calprotectin serially, as we
---

	move towards non-invasive assessments. 10. Please clarify which histologic index will be used? 11. In the endoscopy sub-study - is just the report interpreted centrally, or is there central video recording reviewed by blinded investigators? What does text report and photographic recording mean? 12. In the proposed regression, recommend adjusting for measures of disease severity - albumin, hemoglobin, clinical and endoscopic disease severity
--	--

REVIEWER	Elham Rahme McGill University, Canada
REVIEW RETURNED	02-Jul-2018

GENERAL COMMENTS	This study protocol describes a UK multi-centre (20 sites), two-arm (parallel group), randomised (1:1), placebo-controlled, double-blinded trial including 214 patients that aims to investigate the effect of anakinra (vs. placebo) in addition to IV corticosteroid treatment in patients with acute severe ulcerative colitis (ASUC). The primary outcome is the incidence of medical or surgical rescue therapy (colectomy) within 10 days of IV corticosteroid initiation. The study is funded by Cambridge University Hospitals, National Health Service (NHS) Foundation Trust and University of Cambridge. Ethics approval has been obtained. The protocol is well described in general. However, a few points require clarification. More specifically,  1. Minor: in the Abstract and in the description of the primary outcome: the incidence of medical or surgical rescue therapy (colectomy): add in brackets examples of medical rescue therapies for consistency. 2. Time to clinical response is listed as a secondary outcome. However, it seems that clinical response is measured over the first 5 days. Clarify if this is time to clinical response within the first 5 days of treatment initiation. 3. In Introduction: 'Annual UK hospitalisation rates with acute severe UC (ASUC) are around 3% per annum' say rates among what (the general population, all cause hospitalization, hospitalization for UC patients.. (also no need for 'per annum since it says annual in the beginning). 4. In long-term physical and psychological complications, provide a few examples for physical complications. 5. In 'Those patients who subsequently fail to respond to medical treatment' say over what period of time in general. 6. In 'The barriers for this approach include the toxicity and the financial burden of such treatment', provide a reference and some evidence in support of this argument. For example, are immunosuppressants more toxic than IV corticosteroids? what is the cost of ciclosporin, infliximab and IV corticosteroids for 10 days? 7. 'Anakinra (KineretR), has been used .. with good effect'. Also here provide references and some supportive data. 8. The following statement 'recent research has seen a trend towards antagonism of IL-1 using monoclonal antibodies, anakinra may offer some advantages' is not clear: what is meant by trend toward antagonism of IL-1? what is the role of anakinra in this trend? 9. In 'the cost of the product is much less than that of the
--

	monoclonal antibody alternatives'. Provide the current cost of each for clarity. Is it only because of cost considerations that you chose to study anakinra and not another biologic agent? This is not clear. Maybe you need to add a statement in this regard in the introduction. 10. In Primary Objective: 'To compare the clinical effects of anakinra with placebo..', use a more specific term than clinical effect, as some of the secondary objectives also compare clinical effects. 1. In 'Review of participant outcomes following trial participation may continue for up to 5 years using Hospital Episode Statistics (HES) data, and ' patient readmission and colectomy data (e.g. via HES) may be examined ..', it is not clear if these assessments will take place. Say clearly if they will, otherwise say under what circumstances this follow-up study will take place and provide a short description of the HES data and the population covered. 2. A modified Truelove and Witts severity index (MTWSI)[7] score ≥ 11 is mentioned for the first time in the Inclusion criteria. It needs to be introduced. Say in the introduction what this scale measures and if it has been validated. 3. Secondary Outcomes: again, in 'Time to clinical response (defined as 2nd consecutive day with MTWSI<10)', it is not clear that MTWSI will be measured every day for the 10 + days of the trial. The disease activity mentions measurement over the first 5 days. 4. Also, introduce the scales: EuroQol five dimensions; EQ-5D and Crohn's and ulcerative colitis questionnaire-32; CUCQ-32 and mention their validity. 5. Randomisation will be stratified for previous vs current therapy, it is not clear what the strata are and it is not clear how current is defined. For example if someone was using oral corticosteroid 3 days ago and has stopped, will he/she be considered current or past? List the strata for each stratifying variable and say how they are assessed. Also, discuss if you may have some patients who are neither current nor past oral corticosteroid, for example. 6. In the Method section it is stated that the 'trial will also feature a sub-study specifically examining the endoscopic effects of treatment with anakinra' It is not clear if the sub-study will be conducted only among the anakinra group. Clarify if this is the case or say: 'treatment with anakinra in comparison with placebo' 7. In the statistical section, it is stated that 'the primary analysis will consist of an estimate, of the absolute risk difference of the incidence rates between the two treatment arms, using logistic regression to adjust for important baseline covariates. The logistic regression does not provide absolute risk difference, but odds ratios. This needs to be corrected. Also, you need to justify why you need to adjust for confounders in a randomized, double-blind, clinical trial. 8. In Futility: the statement 'This is equivalent to spending 0.1% from the 15% total beta value set to control the type 2 error under the alternative hypothesis (49% and 29% rates)' is not clear, clarify and provide a reference. 9. 'We will employ the standard approaches to management of missing quality of life data previously described in the CONSTRUCT trial'. Provide examples for clarity.
--	--

VERSION 1 – AUTHOR RESPONSE

IASO Protocol Reviewer Comments

Comment	Response
Editor's Comments	
Please re-locate the first paragraph of your Introduction to either the end of your Introduction or the start of your Methods section	Implemented
Please combine the 'Approvals' and 'Dissemination Plan' sections of your protocol into a single 'Ethics and dissemination' section as per our Instructions for Authors: http://bmjopen.bmj.com/pages/authors/#studyprotocols	Implemented
Dr Peter De Cruz	
The inclusion criteria are all clinically based - there are no endoscopic criteria required for entry.	We thank the reviewer for raising this important point. There has been a move towards use of endoscopic inclusion criteria (or other objective markers of inflammation) for 'explanatory' trials in IBD. In contrast, we have consciously designed IASO as a 'pragmatic' trial. The reasons for this and the implications for trial design are discussed in a new text box in the revised manuscript. In particular, we want to make the trial findings as generaliseable as possible to clinical practice and address whether anakinra should be given alongside IV corticosteroids. Although endoscopic assessment is typically performed during admissions for ASUC, it is not the case that this is typically performed prior to the instigation of corticosteroid therapy, particularly where the clinical evidence for ASUC is strong. For this reason, we have decided to collect endoscopic data for all trial participants where available, and subject this to centralised scoring for use in the trial analyses, but we have explicitly not mandated endoscopic assessment and centralised reading prior to inclusion. It is also important to note the need for centralised scoring would introduce delays into participant screening that would invalidate the

	aim to test anakinra as part of initial therapy for ASUC. In this regard, there are already well established therapies for ASUC when initial therapy with IV corticosteroids has not worked and it is not the aim of IASO to evaluate a new form of medical rescue therapy.
[rescue therapy] assessment is based on the Oxford criteria and the presence of toxic megacolon and clinical judgment. The assessment of these criteria is open to interpretation and may be subject to site and investigator interpretation. Block randomisation by study site may help address this but it is not stated whether block randomisation is part of the randomisation strata	We thank the reviewer for this important point. We did indeed consider stratification by site during randomisation. For a trial with 214 participants we decided to limit the number of strata to be used in block randomisation to 1-2, otherwise the risk of overall imbalance rises too high. Stratification by site is additionally impractical in this regard when the relatively small number of participants are split across 20 sites. Instead, we decided to perform stratification based upon the 2 binary variables identified. We will test for effects of site as part of the analysis.
The rescue rate of 49% in hospitalised patients is quite high (even though it is appreciated that this rate was derived from the CONSTRUCT recruitment data)	The comment from the reviewer is noted. CONSTRUCT provides the best current UK estimate of rates of rescue therapy in this cohort including in many centres that will participate in both studies. We have double checked the details from CONSTRUCT in conjunction with the CONSTRUCT trial manager and trial statistician, and the rescue therapy rate has been confirmed in a nearly identical cohort to that we will recruit for IASO. Nevertheless, the point is well taken and we will pay close attention to rescue therapy rate with the iDMC. It may be possible to reassess trial power in the light of data on rescue therapy available at the interim analysis and consider any necessary steps in conjunction with the trial sponsor and funders.
The drop-out rate may be higher than expected due to the short screening and treatment window as exclusion criteria that are not apparent on screening may develop (eg bacterial gastroenteritis)	Reviewer's comment noted. No text change necessary.
The study would have benefited from video endoscopy as entry and assessment criteria to reduce cohort heterogeneity and improve the accuracy of outcome measure.	Please see comments above and new text box relating to pragmatic trial design.
Whilst there are good mechanistic reasons why anakinra may work in active IBD there appears minimal pilot data to show efficacy. Preliminary data appear limited to a case report (Carter 2003) with worsening of CD and 3 cases of new onset UC after Anakinra (Hugle 2017). Anakinra is well tolerated in published literature; however, high serious infection rates	The reviewer is indeed right that there are no existing robust data to support or reject the use of anakinra in ASUC. We did consider various trial designs that included an open label pilot phase, but rejected these

have been found in RA literature in patients with concomitant steroids. This aspect of the IASO trial will therefore be particularly important to ascertain. Whilst the drug-half life may be short, there may be ongoing effects on the immune system ("immunological half-life") which may interact with infliximab or ciclosporin rescue therapies. The safety analysis of the first 20 patients will therefore be particularly important.	due to the need for a placebo arm to understand the data fully and the relatively large numbers of participants that would be required. Instead, we have adopted a staged approach to a single phase II trial, with feasibility and safety monitoring, precisely along the lines suggested by the reviewer, as well as an interim futility analysis. If there were concerning signals of safety and/or futility, of the sort the reviewer identifies, we are confident that our trial design and monitoring arrangements would detect this and lead to early trial termination.
Siddharth Singh	
In the abstract, please clarify what phase trial is this? PICOs need to be more clearly defined. Please also clarify that the intention is to use Anakinra for a short duration.	Implemented
Please detail out MTWS index - is it a validated index for UC? Suggest using alternative indices too to facilitate comparison across trials such as the Lichtiger index?	We have added further details concerning the MTWSI as an online appendix to the manuscript along with further details outlining our selection of this metric. The reviewer is, of course, correct to highlight the lack of any validated clinical index of severity in ASUC. The MTWSI is an alternative name for the Lichtiger index, as suggested by the reviewer. This index is easy to determine and has been used in a number of recent clinical trials in this area. Importantly, this includes the index trial of ciclosporin as rescue therapy, as well as in the CySIF and CONSTRUCT trials comparing ciclosporin and infliximab, with an equivalent inclusion cut-off. The components of the MTWSI include those parameters that are relevant to clinical decision making for initial recognition of ASUC and for determination of the need for rescue therapy.
Is endoscopic confirmation of disease activity a requirement for inclusion, for all patients?	Endoscopic confirmation is not a requirement for trial inclusion. The trial has been designed to be as pragmatic as possible and therefore includes patients where the investigator suspects that ASUC is the most-likely diagnosis. We have further elaborated regarding this decision in the dedicated text box.
What tests would be undertaken to rule out infection - GI pathogen panels based on nucleic acid amplification have 30% positivity, as compared conventional stool cultures? What is considered a "significant GI pathogen"	No specific tests are mandated for GI pathogen detection and sites are able to implement local standard practice. PCR based testing is not widely available within NHS labs at present. Centres are free to use whatever means they have to test and confirm pathogens coupled

	with local physicians' assessment for significance
I think it would be helpful to standardize IV corticosteroid course across centers to allow results to be interpretable	This is a pragmatic study and we are keen not to alter care pathways that are in place in hospitals. The NHS care pathway permits Trusts to select either hydrocortisone OR methylprednisolone as their initial intravenous corticosteroid of choice. We will analyse for differences between patients receiving these steroid regimens across the trial (as well as in the placebo arm alone).
I am unsure about including patients who present with newly diagnosed UC patients treated with oral corticosteroids - what is proof of failing outpatient steroids?	In line with the pragmatic approach to the trial, we are attempting to maximise the real-world applicability of early anakinra care in ASUC. Accordingly, we are permitting trial enrolment for all patients meeting the inclusion criteria based upon stool frequency and other MTWSI criteria, and a need for admission in the judgement of the treating clinician. We acknowledge that this approach introduces heterogeneity but it also maximises the applicability of any results. This also makes trial recruitment feasible given rarity of this presentation.
I'm unsure but is concomitant ciclosporine allowed??	Currently, the protocol would indeed allow patient receiving ciclosporin to be enrolled in the trial. While we consider it unlikely that patients will present with concomitant ciclosporin we do acknowledge that this presents an imbalance vs anti-TNFs in the exclusion criteria. Accordingly, we will incorporate a formal change to the exclusion criteria into a future substantial amendment to redress this balance. We would like to thank the reviewer for pointing this out to us.
If the goal is to understand 3 month outcomes, then I strongly suggest ascertaining treatment patterns after discharge, besides colectomy? When will physicians be unblinded? Will that influence subsequent therapy?	Hospital admissions will be tracked by means of HES at 1 year post randomisation and up to 5 years post randomisation. Furthermore, quality of life measures are also being monitored at baseline, 3 months and 6 months. Physicians will not be unblinded prior to full data analysis unless for valid medical reasons. Accordingly, subsequent therapy should not be influenced for several years if at all following IMP treatment.
If possible, suggest measuring fecal calprotectin serially, as we move towards non-invasive assessments	Stool sample collection at Day 0 and Day 5 will be used for assaying faecal calprotectin in addition to GI pathogens
Please clarify which histologic index will be used?	Due to the dynamic nature of the field, with the potential for newer indices to become validated, the choice of specific histologic index has

	been left open. On current evidence, we are planning to use the Geboes Index. However, we also recognise that emerging indices such as Nancy or Robarts may be available and more suitable based on evidence available in the future. The selection of the most appropriate index by the trial management group will be aided by seeking expert histopathology advice nearer the end of the trial.
In the endoscopy sub-study - is just the report interpreted centrally, or is there central video recording reviewed by blinded investigators? What does text report and photographic recording mean?	Both the reports and the images from the sub-study will be reviewed and assessed by local and central blinded investigators. In some sites video will not be available and static images will be reviewed. The wording in the manuscript has been updated to provide more clarity related to the terms
In the proposed regression, recommend adjusting for measures of disease severity - albumin, hemoglobin, clinical and endoscopic disease severity	These will be included in the SAP as a secondary analysis. We restrict the description to just the primary analysis in the protocol.
Elham Rahme	
Minor: in the Abstract and in the description of the primary outcome: the incidence of medical or surgical rescue therapy (colectomy): add in brackets examples of medical rescue therapies for consistency	Implemented
Time to clinical response is listed as a secondary outcome. However, it seems that clinical response is measured over the first 5 days. Clarify if this is time to clinical response within the first 5 days of treatment initiation.	Implemented
In Introduction: 'Annual UK hospitalisation rates with acute severe UC (ASUC) are around 3% per annum' say rates among what (the general population, all cause hospitalization, hospitalization for UC patients.. (also no need for 'per annum since it says annual in the beginning).	Implemented
In long-term physical and psychological complications, provide a few examples for physical complications	Examples have now been provided in the main text
In 'Those patients who subsequently fail to respond to medical treatment' say over what period of time in general.	This has now been addressed in the text (typically 4-10 days)
In 'The barriers for this approach include the toxicity and the financial burden of such treatment', provide a reference and some evidence in support of this argument. For example, are	References have been added to the text.

CCTU/776/020 v7 27/03/2018

immunosuppressants more toxic than IV corticosteroids? what is the cost of ciclosporin, infliximab and IV corticosteroids for 10 days?	
Anakinra (KineretR), has been used .. with good effect'. Also here provide references and some supportive data.	The requested evidence has been cited
The following statement 'recent research has seen a trend towards antagonism of IL-1 using monoclonal antibodies, anakinra may offer some advantages' is not clear: what is meant by trend toward antagonism of IL-1? what is the role of anakinra in this trend?	A citation has been added to demonstrate the variety of therapeutic investigations, including monoclonal antibody research targeting IL-1.
In 'the cost of the product is much less than that of the monoclonal antibody alternatives'. Provide the current cost of each for clarity. Is it only because of cost considerations that you chose to study anakinra and not another biologic agent? This is not clear. Maybe you need to add a statement in this regard in the introduction.	Citations for the costs have been added. The additional justifications for using anakinra (e.g. half-life, non-specificity for IL-1 subtype) are already discussed in the paragraph in question.
In Primary Objective: 'To compare the clinical effects of anakinra with placebo..', use a more specific term than clinical effect, as some of the secondary objectives also compare clinical effects.	We are unable to change this wording as this is identical to the wording used in the approval protocol for the trial which is currently in active use.
In 'Review of participant outcomes following trial participation may continue for up to 5 years using Hospital Episode Statistics (HES) data, and ' patient readmission and colectomy data (e.g. via HES) may be examined ..', it is not clear if these assessments will take place. Say clearly if they will, otherwise say under what circumstances this follow-up study will take place and provide a short description of the HES data and the population covered.	Additional text has been added from the full protocol to explain the timing and duration of the HES assessments. The HES data collection that is already planned (patient readmissions and colectomy rates) has already been specified in the text.
A modified Truelove and Witts severity index (MTWSI)[7] score ≥ 11 is mentioned for the first time in the Inclusion criteria. It needs to be introduced. Say in the introduction what this scale measures and if it has been validated.	This has been introduced more thoroughly in the main body. Additionally, we have added the MTWSI score sheet for use in the trial as an appendix
Secondary Outcomes: again, in 'Time to clinical response (defined as 2nd consecutive day with MTWSI<10)', it is not clear that MTWSI will be measured every day for the 10 + days of the trial. The disease activity mentions measurement over the first 5 days.	Time to clinical response will be measured during the treatment phase only. This has been clarified in the text.

Also, introduce the scales: EuroQol five dimensions; EQ-5D and Crohn's and ulcerative colitis questionnaire-32; CUCQ-32 and mention their validity.	Some introductory text and references related to the questionnaires' validation have been added
Randomisation will be stratified for previous vs current therapy, it is not clear what the strata are and it is not clear how current is defined. For example if someone was using oral corticosteroid 3 days ago and has stopped, will he/she be considered current or past? List the strata for each stratifying variable and say how they are assessed. Also, discuss if you may have some patients who are neither current nor past oral corticosteroid, for example.	The "vs" was a typo which has now been corrected to "or". We thank the reviewer for bringing this to our attention. The correction should hopefully now make the section self-explanatory. However, in case more clarification is needed; participants will be grouped by yes vs no answers to the stratification variables. Accordingly, using the 2 scenarios put forward by the reviewer  1. The first participant would be "yes" for current or previous oral corticosteroid prescription(s) within 8 weeks prior to first dose of intravenous corticosteroids 2. We do indeed expect to have some participants who have neither current nor past oral corticosteroid treatment and these would fall into the "no" group for this stratification variable. As the reviewer points out, this binary classification will introduce some situations where biologically a patient may belong more to the other group (such as in the instance the reviewer describes, which is relatively rare) but we considered this to be the simplest and most relevant classification to apply to the sorts of patients that are admitted for ASUC.
In the Method section it is stated that the 'trial will also feature a sub-study specifically examining the endoscopic effects of treatment with anakinra' It is not clear if the sub-study will be conducted only among the anakinra group. Clarify if this is the case or say: 'treatment with anakinra in comparison with placebo'	Implemented; the substudy will feature participants from both treatment arms
In the statistical section, it is stated that 'the primary analysis will consist of an estimate, of the absolute risk difference of the incidence rates between the two treatment arms, using logistic regression to adjust for important baseline covariates. The logistic regression does not provide absolute risk difference, but odds ratios. This needs to be corrected. Also, you need to justify why you need to adjust for confounders in a randomized,	The analysis will be using the generalised linear model framework where the endpoint is a binary variable, but using an identity link function. Hence the description is "logistic regression". As per CONSORT guidelines we are analysing the data in line with the randomisation process, which is stratified. Hence the analysis must adjust for randomisation strata. Further baseline covariates are pre-identified as influential confounders, hence adjusting for them will

double-blind, clinical trial.	increase the accuracy of the treatment effect estimate.
In Futility: the statement 'This is equivalent to spending 0.1% from the 15% total beta value set to control the type 2 error under the alternative hypothesis (49% and 29% rates)' is not clear, clarify and provide a reference.	A reference to the DeMets & Lan error spending paper has been added.
'We will employ the standard approaches to management of missing quality of life data previously described in the CONSTRUCT trial'. Provide examples for clarity.	A an example from CONSTRUCT has been added. The full description and further actions can be found in the citation.

VERSION 2 – REVIEW

REVIEWER	Siddharth Singh University of California San Diego, USA
REVIEW RETURNED	09-Oct-2018

GENERAL COMMENTS	Thank you for addressing all comments. Generally pragmatic trials are phase IV studies performed for treatment strategies with approved medications. For a drug like Anakinra which has minimal pilot data in IBD, I do not think a pragmatic trial as designed will be the "definitive" trial to change treatment. I recognize challenges with ASUC trials, but as we have learned from some promising but failed trials in IBD, standardization of patient enrollment and outcome assessment is critical to getting important information from a phase 2 trial.
---